# Growth after Trauma: The Role of Self-Compassion following Hurricane Harvey

Joshua Yuhan [1,*], David C. Wang [1] , Andrea Canada [1] and Jonathan Schwartz [2]

[1] Rosemead School of Psychology, Biola University, La Mirada, CA 90639, USA; david.wang@biola.edu (D.C.W.); andrea.canada@biola.edu (A.C.)
[2] College of Public Service, University of Houston-Downtown, Houston, TX 77002, USA; schwartzj@uhd.edu
* Correspondence: joshua.w.yuhan@biola.edu

**Abstract:** The psychological impact of a traumatic event includes potentially both negative (e.g., PTSD, depression, and anxiety) as well as positive (e.g., post-traumatic growth) outcomes. The construct of self-compassion—the capacity to be compassionate towards oneself—has been associated with various psychological benefits following disasters; however, the association between self-compassion and PTG have not yet been examined in natural disaster settings. This study aimed to examine the relationship between these constructs, with self-compassion as a potential mediator in this relationship. Three hundred and nine undergraduate students affected by the impact of Hurricane Harvey were recruited. Statistical analyses revealed a significant mediation effect, with PTSD symptoms being both directly and indirectly (via self-compassion) associated with PTG. The capacity to grow from traumatic experiences is mediated by one's disposition to be compassionate towards oneself, serving as a resilience factor to provide individuals with the cognitive and emotional resources to grow after trauma. These findings have significant implications in both clinical and research contexts, including the use of self-compassion interventions to protect against PTSD and other comorbid psychopathology and also act as a catalyst for growth following natural disaster events.

**Keywords:** natural disaster trauma; post-traumatic growth; self-compassion; PTSD symptoms

## 1. Introduction

The experience of traumatic events can have a significant impact on an individual. Trauma represents an accumulation of many factors that manifests itself in different ways; therefore, it is complicated to understand [1]. Individuals who have experienced trauma may develop psychological distress which can lead to long-term implications negative to health and well-being. It is therefore of great importance to understand vulnerability and resilience factors in relation to traumatic events. Research has now developed a new direction of looking at the dose–response relationship between traumatic stressors and the degree of psychopathology that may follow. Natural disaster studies provide a valuable opportunity to look at the stressor gradient in a uniform fashion in order to better understand trauma and its underlying mechanisms [2]. Consequently, there has been a growing body of literature on the effects of natural disaster trauma on individuals.

### 1.1. Natural Disaster Trauma

Natural disasters include all types of severe weather disturbances which pose a threat to human health and safety. Since 1995, over 4.4 billion people have been affected by natural disasters, causing over USD 2 trillion in economic losses [3]. Through history, one can understand the impact that natural disasters may have on an individual. The June 2008 Midwest floods affected over 11 million people while Hurricane Katrina in 2005 led to over USD 125 billion in damages, evacuations, and lost property. These environmental events are random and unpredictable; however, they are deemed "natural disasters" dependent

on the relative negative impact on human life. Individuals affected by natural disasters face psychological repercussions that can have serious and long-term emotional ramifications.

The exposure to prolonged and potentially traumatizing events were on full display during Hurricane Harvey. During the five days, Hurricane Harvey dropped more than 33 trillion gallons of rain and set a continental US record for rainfall at 51.88 inches [4]. Among 13 million people who were directly affected by the storm, more than 22,000 individuals were rescued from the floods, an estimated 32,000 displaced survivors were temporarily housed in shelters, and more than 100,000 homes were damaged. Disasters and their aftermath are sources of enormous psychological stress and are likely to disrupt the lives of not only those directly affected, but also emergency first-responders, and tertiary victims as well.

While acute stress reactions are normative and typically follow a traumatic event, PTSD develops when pathological symptoms seem to persist and increase in severity. Symptoms of PTSD (e.g., intrusive thoughts, avoidance, and hyperarousal) have been significantly correlated with higher PTSD prevalence and lower levels of psychological functioning and quality of life [5,6]. Additionally, increased traumatic symptoms are also related to increases in other comorbid psychopathology including depression, generalized anxiety disorder (GAD), and other psychiatric disorders [7].

### 1.2. Post-Traumatic Growth

Focusing on research pertaining to solely negative aspects of trauma and psychopathology can lead to a potentially incomplete understanding of the potential psychological impact of trauma. Though counterintuitive, researchers have theorized that positive outcomes can also come out of traumatic events [8]. More specifically, studies on post-traumatic growth (PTG) captures responses to trauma in which positive change is directly experienced as a result of coping with traumatic events [9]. For example, in the aftermath of trauma, people have been found to gain a greater appreciation toward life, become more in touch with their spirituality, and develop more intimate relationships with others. PTG is understood to be achieved through cognitive rumination coupled with the ability to process the emotional experiences of the trauma. PTG has also been displayed as a self-regulation mechanism, which helps protect oneself from psychological distress [10].

The PTG model created by Tedeschi and Calhoun (1996) assumes that in order for growth to occur after trauma, some degree of psychological distress is necessary. However, a large percentage of individuals have been found to be relatively unaffected by traumatic events [11]. Of those affected, individuals may simply fail to perceive the event as a crisis and therefore they have little reason for either distress or growth. Others experience a combination of both positive and negative outcomes. The literature has identified that there are a range of responses to trauma: PTG may be positive [12,13], negative [14–16], or even insignificant [11,17]. Understandably, this has led to further interest in the relationship between psychological distress and PTG.

### 1.3. Self-Compassion

According to Neff (2003), self-compassion is defined as a kindness towards oneself during times of trials [18]. This construct has been categorized into three key tenets: self-kindness, common humanity, and mindfulness [19]. The first component of self-kindness refers to the tendency to be caring and understanding towards oneself during difficult times. The second component of common humanity is concerned with the inclination to view the traumatic events as a normal part of human life, a connection to the overall human experience. The third and final component, mindfulness, refers to the ability of being aware of difficulties, without becoming too absorbed or over-identifying with the negative and maladaptive emotions. Consequently, self-compassion has been theorized as a resilience factor to promote growth and protect against the onset of psychopathology.

Research has found that self-compassion is positively related to psychological well-being and is positively associated with life satisfaction and social connectedness in healthy

adult populations [20]. Conversely, self-compassion is also negatively related to depressive and anxiety symptoms in clinical samples [21] and decreased levels of cognitive rumination, thought suppression, and self-criticism [18]. These extensive findings pertaining to the utility of self-compassion has ultimately led to the development of self-compassion programs as a general therapeutic approach serving as a form of "self" social support [22].

There has been growing research interest in exploring and understanding the possibility that compassion for others might be enhanced in an individual post-trauma [23]; however, to our knowledge, few studies have thus far explored the relationship between self-compassion and PTG. One study measured the relationship between self-compassion and PTG in a sample of 601 college students who reported having experienced at least one crisis or negative life event [24]. In this study, the authors did not find a significant direct relationship between self-compassion and PTG. However, the researchers tested a mediation model using structural equation modeling and found significant indirect effects of self-compassion on PTG through positive reframing. Despite these findings, the question of whether and to what extent self-compassion is directly related to PTG is largely unknown. Accordingly, future research regarding this potential direct association represents a significant gap within the trauma literature.

Neff (2003) provided an explanation of self-compassion as identifying one's own events (whether they are painful or not) as a function of the broader human experience, knowing that we are not alone in our suffering and negative experiences [18]. To better understand the relationship between psychological distress, self-compassion, and PTG, researchers have also investigated possible moderator effects. For example, one study found that the emotional response of loneliness significantly moderated the association between low levels of social support and the onset of psychopathology in Israeli soldiers in the Lebanon War [25]. Additionally, a study by Zeligman, Bialo, Brack, and Kearney (2017) reported loneliness as a moderating factor and a potential barrier to trauma recovery [26].

This validating disposition associated with self-compassion sharply contrasts with the isolating and potentially shaming effects of loneliness. Building upon this contrast, might it be possible that self-compassion functions as a resiliency factor for trauma survivors just as loneliness functions as a risk factor? Additionally, just as avoidant coping mechanisms and self-blame are linked to higher psychological distress [27], might the practice of self-compassion—which leads one to mindfully approach one's struggles with kindness—represent a key factor that accounts for the relationship between trauma and its potential positive and negative sequalae?

Accordingly, the purpose of this study was to empirically investigate the relationship between PTSD symptoms, PTG, and self-compassion—with self-compassion hypothesized to mediate the relationship between PTSD symptoms and PTG. Based on previous research, we believe that one's disposition to be compassionate towards oneself will represent a key ingredient that enables a trauma survivor to grow from their traumatic experience.

## 2. Materials and Methods

### 2.1. Participants

The sample consisted of 309 participants recruited from a convenience sample of students enrolled at a large public university in Southwest Texas. Inclusion criteria for this study required that participants must have lived in an area directly impacted by Hurricane Harvey. The selection of participants focused on individuals over 18 years of age who were asked to describe the extent to which their lives have been personally affected by Hurricane Harvey. Participants were compensated with course credit for their time. The mean years of age of the sample was 21.89 with a standard deviation of 4.33. The racial majority of the sample was Hispanic (36%), followed by Asian (24.2%), White (22.8%), Black (11.6%), Multiracial (2.9%), and Other (1.9%). The sample population reflects the University of Houston student body relating to gender (73.9% Female, 26.1% Male).

## 2.2. Measures

PTSD Symptoms. PTSD symptoms were measured by the post-traumatic stress disorder checklist (PCL-5). The PCL-5 is a self-reported measure of PTSD symptoms developed by Blevins, Weathers, Davis, Witte, and Domino [28]. In addition, the current version of the PCL-5 reflects revisions found in the Diagnostic and Statistical Manual of Mental Disorders [6]. The 20-item measure describes PTSD symptoms in the last month across the main cluster criteria of PTSD (i.e., intrusive memories, avoidance, and negative alterations in arousal, mood, and cognition) based on a 0 to 4-point Likert scale (0 = not at all and 4 = extremely). Total scores ranged from 0 to 40, with higher scores reflecting greater symptomatology. For our present sample, this measure demonstrated high internal reliability ($\alpha = 0.94$).

Self-Compassion. Self-compassion was measured by the short form of the self-compassion scale (SCS-SF). The SCS-SF is a 12-item self-report scale, which measures the degree to which an individual displays kindness towards oneself, generally referred to as self-compassion [29]. The SCS-SF is divided into three components: Self-kindness, Common Humanity, and Mindfulness. For our present sample, the overall internal reliability of this scale was reported as good ($\alpha = 0.71$).

Post-Traumatic Growth. PTG was measured by the short form of the post-traumatic growth inventory (PTGI-SF). The PTGI-SF is an 11-item self-report measure used to assess the perceived changes following a traumatic event [30]. Participants responded using a 1 to 7-point Likert scale (1 = not at all and 7 = to a very great degree) with total scores between 11 and 77. For our present sample, the overall internal reliability of this scale was reported as excellent ($\alpha = 0.92$).

## 2.3. Procedures

In exchange for credit toward an undergraduate psychology course, participants were asked to fully complete an online questionnaire related to one's experiences following Hurricane Harvey. The multi-section questionnaire consisted of three total sections measuring PTSD symptoms, self-compassion, and post-traumatic growth (i.e., PCL-5, SCS-SF, and PTGI-SF), along with a demographic questionnaire and descriptions of how participants have been personally affected by Hurricane Harvey. Participants were instructed to fill out the multi-section questionnaire as accurately as possible; time of completion for participants averaged approximately 30 min.

## 2.4. Data Analysis

Mediation analyses were tested through the Andrew Hayes PROCESS Macro version 2.6, which allows for "conducting mediation, moderation, or conditional process analysis..." rather than utilizing multiple tools, such as INDIRECT, SOBEL, and MODMED [31]. For our analysis, we utilized Model 4, which tests both the direct and indirect effects (through mediation) of an independent variable on a dependent variable. For the purposes of our experiment, PTSD symptom count was identified as the independent variable, self-compassion was entered as the mediator variable, and PTG was entered as the dependent variable. Age, gender, ethnicity, and the severity of the trauma were included in the model as covariates to evaluate the unique association of self-compassion above and beyond the association with the other covariates. The first step of PROCESS Macro tested the association between PTSD symptoms and PTG when all demographic variables and covariates are controlled for (path *a*). Next, it was important to identify the coefficient for self-compassion as a predictor of post-traumatic growth (path *b*). Third, the direct effect of PTSD symptoms as a predictor of post-traumatic growth, when all variables are controlled for, was established (path *c'*). Fourth, the total indirect effect (path *c*) was found, through the product (interaction) of path *a* and path *b* coefficients.

## 3. Results

### 3.1. Preliminary Analyses

Before analyzing the main hypotheses, preliminary analyses were conducted to address missing data, blatant response sets, and outliers. Of the original 320 participants of the study, 4 were excluded due to their response sets having more than 10% of items missing. After accounting for unusually short survey completion times and searching for discernable response patterns, we excluded seven more response sets to be omitted due to blatant random response patterns and a completion time of less than five minutes. As such, the final number of cases analyzed was 309.

Next, descriptive statistics of the major constructs in the study were calculated (see Table 1). On average, participants reported moderate levels of PTSD symptoms, moderately high levels of self-compassion scores, and high levels of PTG scores. Analyses examining the normality of the data indicated that all variables were within acceptable ranges for skewness and kurtosis within the acceptable range between $-3$ and $3$ (see Table 1).

**Table 1.** Descriptive statistics of study variables: PTSD symptoms, self-compassion, and PTG.

| Variable | Range | | | | | | |
| | *M* | *SD* | Possible | Actual | Skew | Kurts | Alpha |
|---|---|---|---|---|---|---|---|
| PTSD Symptoms (PCL-5) | 16.23 | 7.93 | 8–40 | 8–40 | 0.819 | −0.172 | 0.94 |
| Self-Compassion (SCS-SF) | 37.20 | 7.07 | 12–60 | 19–60 | 0.591 | 0.854 | 0.71 |
| Post-Traumatic Growth (PTGI-SF) | 60.17 | 11.36 | 11–77 | 26–77 | −0.362 | −0.671 | 0.92 |

### 3.2. Mediation Analysis

Hayes's (2013) PROCESS Macro 2.16 was used to examine the indirect influence of PTSD symptoms on PTG through self-compassion (see Tables 2 and 3). Using an ordinary least squares regression-based path analysis, it was found that PTSD symptoms indirectly influenced PTG through its relationship with the mediator, self-compassion. Table 2 illustrates that individuals with higher levels of PTSD symptoms had lower levels of self-compassion ($a = -0.21$, $p < 0.001$), and those who had higher levels of self-compassion had more PTG ($b = 0.55$, $p < 0.001$). Tables 2 and 3 illustrate that the total effect of PTSD symptoms on PTG ($c = -0.28$, $p < 0.001$) decreased when the mediator, self-compassion, was controlled for ($c' = -0.16$, $p = 0.040$). A 95% bias-corrected bootstrap confidence interval for the indirect effect of self-compassion of PTG based on 1000 bootstrap samples did not include zero ($-0.17$ to $-0.06$), and the Sobel test (normal theory test) also indicated that the mediation effect was significant ($Z = -3.31$, $p < 0.001$). Therefore, the results provide evidence that self-compassion partially mediated the influence that PTSD symptoms have on PTG.

**Table 2.** Model coefficients for the simple mediation model for PTSD symptoms, self-compassion, and PTG (Path *a*, *b*, *c'*).

| | Consequent Variable | | | | | |
| | M (Self-Compassion) | | | Y (PTG) | | |
| Covariates | Coeff. | *SE* | *p* | Coeff. | *SE* | *p* |
|---|---|---|---|---|---|---|
| Age | 0.01 | 0.01 | 0.702 | −0.01 | 0.01 | 0.061 |
| Gender | 0.13 | 0.89 | 0.882 | 3.50 | 1.33 | 0.009 |
| Ethnicity | −0.28 | 0.33 | 0.389 | 0.28 | 0.49 | 0.565 |
| Trauma Severity | −0.21 | 0.13 | 0.103 | 0.85 | 0.19 | 0.000 |
| Antecedent Variable | | | | | | |

**Table 2.** *Cont.*

| | | Consequent Variable | | | | | | |
|---|---|---|---|---|---|---|---|---|
| | | M (Self-Compassion) | | | | Y (PTG) | | |
| **Covariates** | | **Coeff.** | *SE* | *p* | | **Coeff.** | *SE* | *p* |
| X (PTSD symptoms) | *a* | −0.21 | 0.05 | 0.000 | *c′* | −0.16 | 0.08 | 0.040 |
| M (self-compassion) | $i_1$ | - | - | - | *b* | 0.55 | 0.08 | 0.000 |
| Constant | | 42.15 | 2.15 | 0.000 | $i_2$ | 31.63 | 4.84 | 0.000 |
| | | $R^2 = 0.07$ | | | | $R^2 = 0.20$ | | |
| | | $F(5, 299) = 4.94$ | | | | $F(6, 298) = 12.53$ | | |
| | | $p < 0.001$ | | | | $p < 0.001$ | | |

*Note.* $N = 309$. *SE* = standard error. Coeff = unstandardized coefficient. X = independent variable (PTSD symptoms). *a* = path *a*; the association between the independent variable (PTSD symptoms) and mediator (self-compassion). *b* = path *b*; the association between the mediator (self-compassion) and dependent variable (PTG). *c′* = path *c′*; the direct effect of the independent variable (PTSD symptoms) on the dependent variable (PTG). $i_1$ = the intercept, or constant, for the association between PTSD symptoms and self-compassion. $i_2$ = the intercept, or constant for the association between self-compassion and PTG.

**Table 3.** Model coefficients for the total effect of PTSD symptoms on PTG (Path *c*).

| **Covariates** | | **Coeff.** | *SE* | *p* |
|---|---|---|---|---|
| Age | | −0.01 | 0.01 | 0.103 |
| Gender | | 3.57 | 1.43 | 0.013 |
| Ethnicity | | 0.12 | 0.52 | 0.810 |
| Trauma Severity | | 0.73 | 0.21 | 0.000 |
| **Antecedent Variable** | | Y (PTG) | | |
| X (PTSD symptoms) | *c* | −0.28 | 0.08 | 0.000 |
| | $i_2$ | 55.03 | 3.42 | 0.000 |
| | | $R^2 = 0.09$ | | |
| | | $F(5, 299) = 5.91$ | | |
| | | $p < 0.001$ | | |

*Note.* $N = 309$. *SE* = standard error. Coeff = unstandardized coefficient. X = independent variable (PTSD symptoms). *c* = path *c*; the total effect of the independent variable (PTSD symptoms) on the dependent variable (PTG). $i_2$ = the intercept, or constant, for the association between self-compassion and PTG.

## 4. Discussion

The aim of the current investigation was to test the direct and indirect effects (via self-compassion) of PTSD symptoms on PTG. First, the means for the PCL-5 illustrated that our sample tended to report moderate levels of PTSD symptoms ($M = 16.23$, $SD = 7.93$). This demonstrates that individuals in the current sample experienced a variety of PTSD symptoms and severity, ranging from scores of 8 to 40. A score of 19 on the PCL-5 corresponded to a sensitivity of 0.83, 95% CI (0.67, 0.92) and a specificity of 0.39, 95% CI (0.29, 0.50) for a PTSD diagnosis [32]. The PCL-5 has been previously utilized as a screener for PTSD and to determine symptom severity, as the cut score of 19 resulted in a PPV of 37.8% and a NPV of 83.7%. The means for the SCS-SF and PTG-I also displayed moderately high levels of self-compassion scores ($M = 37.20$, $SD = 7.07$) and high levels of PTG scores ($M = 60.17$, $SD = 11.36$) which indicated that individuals in this sample population experienced, on average, robust characteristics of self-compassion and PTG after Hurricane Harvey.

Furthermore, mediation analyses supported our hypotheses that PTSD symptoms would have a significant impact on PTG both directly and indirectly through the mediator, self-compassion. Beginning with the significant direct effects observed in our study (see Table 4), the analysis showed that there was a significant negative association between PTSD symptoms and PTG [12]. It seems that the traumatic event experienced through the natural disaster may lead to a wide range of physical and mental health complications that threaten the physical and psychological equilibrium of an individual. Those more severely

affected by the natural disaster may be unable to grow from the subsequent trauma. Thus, an individual's ability for PTG may be hindered and serve as a barrier towards growth.

**Table 4.** Zero-order correlations: PTSD symptoms, self-compassion, and PTG.

|  |  | 1 | 2 | 3 |
|---|---|---|---|---|
| 1 | PTSD Symptoms (PCL-5) | 1 |  |  |
| 2 | Self-Compassion (SCS-SF) | −0.260 ** | 1 |  |
| 3 | Post-Traumatic Growth (PTGI-SF) | −0.176 ** | 0.344 ** | 1 |

*Note.* ** Correlation is significant at the 0.01 level (2-tailed).

Further, the analysis indicated that PTSD symptoms were significantly and negatively associated with self-compassion. This is consistent with previous literature which indicates that PTSD symptoms are negatively correlated with self-compassion [12,33]. In fact, high levels of self-compassion have been inversely correlated with all symptom clusters for the DSM-V [34]. As seen in the Diagnostic and Statistical Manual of Mental Disorders [6], cognitive distortions and negative beliefs can be a negative consequence and symptom of PTSD. For those with increased PTSD symptoms due to trauma, there may be an internalization of self-blame [27]. After a natural disaster, one may begin to question and wonder if there was anything that could have been done differently. This can create an illusory depiction of control as a person desires to understand why the disastrous event has occurred, leading to the belief that the disruptive effects of the natural disaster could have been stopped, altered, or reduced in some way. Rather than understanding that the trauma (i.e., natural disaster) was an act of nature outside one's control, an individual may resist viewing the disaster through a self-compassionate lens: unable to reduce self-judgment, isolation, and the rumination of negative thoughts and feelings [18]. Altogether, these tendencies are a barrier to adaptive cognitive processing of the traumatic event and hinder accurate self-appraisal—a disposition salient to successfully navigating natural disasters [35]. Subsequently, a person may be unable to display self-compassion and continue to blame oneself for the incident, believing that one's actions played a role in the natural disaster. There is a lack of literature looking at the negative effects of cognitive rumination and how self-blame may hinder a person's ability to display self-compassion. Further research is needed to identify these effects.

Importantly, the analysis displayed a strong positive association between self-compassion and PTG. The zero-order correlation displayed a significant positive relationship between self-compassion and PTG; however, our results through Hayes Mediation Analyses supported an even stronger association. Wong and Yeung (2017) identified a significant indirect effect of self-compassion on PTG through positive reframing [24]; however, our study is one of the first to validate the direct effects of self-compassion on PTG in the trauma literature. It seems that the core values of self-compassion (i.e., self-kindness, common humanity, and mindfulness) play an integral role in understanding growth after trauma. This research is the first to reveal such a strong and direct association between self-compassion and PTG in the natural disaster trauma literature.

Additionally, results from our study indicate that self-compassion significantly mediated the relationship between PTSD and PTG. While previous literature has looked at the mediation of loneliness [36], no research to our knowledge has investigated self-compassion as a mediator between PTSD and PTG. Self-compassion can further explain this relationship by looking at the more positive outlooks on trauma in which self-compassion addresses the individual's perceived connection with common humanity and decreases in self-blame [33]. Often conceptualized as the experience of supporting oneself during difficult times, self-compassion may serve as a resilience factor which enables an individual to have the cognitive and emotional resources to grow after trauma [37]. It is important to stress that the present findings do not imply that self-compassion is the only mediator of the relationship between PTSD symptoms and PTG. Future research can look to identify other possible mediators of this relationship to better understand the development

of growth after trauma. For example, constructs pertaining to an individual's religion and/or spirituality may be a direction worth exploring, considering its relevance not only to trauma and trauma recovery [38,39], but also to one's disposition to be compassionate towards oneself [40].

### 4.1. Implications for Future Research and Practice

These results have important implications in assisting individuals after a traumatic experience, especially in the natural disaster literature. Specifically, for those involved in clinical work, this research indicates that the virtue of self-compassion may be beneficial for growth after trauma. Creating a self-compassionate frame of mind in the face of trauma can serve as an important source of resilience and future well-being [18,20].

Growing research on self-compassion has found important therapeutic implications for treatment. However, there is a lack of current trauma interventions that explicitly tap into the virtue of self-compassion. Various therapies including Acceptance and Commitment Therapy (ACT) [41] and Compassion-Focused Therapy (CFT) [42] promote self-compassion by looking at the mechanisms of self-blame and criticism. Given the benefits of self-compassion and its effects on PTG, there needs to be increased focused on how one may teach others to become more self-compassionate in natural disaster settings. Short-term writing interventions and brief self-compassion training programs have also been found to be helpful to develop a self-compassionate mindset [43–45].

There is a need to translate these types of interventions for individuals with PTSD in natural disaster settings. The treatment programs may consist of aspects of acceptance, common humanity (broader experience), the reduction of self-blame, and the increase of self-kindness. Many of the current interventions focus on the aspect of mindfulness as it aims to decrease maladaptive emotions and negative experiences [42]. It is important to identify these maladaptive coping strategies that target core beliefs of shame and guilt. From a cognitive-behavioral perspective, self-compassion can be conceptualized as an adaptive coping mechanism that increases positive thoughts and decreases avoidance-oriented coping strategies [33,46]. From the results of this study, the embodiment of self-compassion as a reduction of self-judgment, isolation, and negative rumination can be further extended to a natural disaster trauma-exposed population.

Many individuals feel responsible for the negative consequences and aftermath of a natural disaster [47]. A sense of control may be developed as a means of coping with the trauma. With the knowledge of a broader human experience and the introduction of self-kindness, an individual may develop the means to grow from pre-existing trauma. Recognizing that the human condition is imperfect, and that we are not alone in our suffering, individuals affected by natural disasters may hope to find some solace and resolve in the relational nature of the traumatic event [18]. In many cases, natural disasters bring communities together, and develop even stronger bonds and identities [48]. Individuals thus discover that they are not alone. A warm and self-compassionate identity may be a key indicator for growth after trauma, as external relationships model appropriate behaviors and thoughts of acceptance and change. Therefore, newer forms of self-compassion trainings specifically related to natural disaster settings and PTSD should be investigated.

### 4.2. Limitations

While these results contribute to the understanding of some important factors regarding growth following traumatic experiences, there are some limitations. The current study was limited to a convenience sample of undergraduate students from a public university in Southeastern Texas. Subsequently, this sample may represent a socio-economic status, education status, and age range that is not representative of a larger population. It would be important to replicate these findings with a more generalizable natural disaster population. Another limitation of the study was the use of brief self-report measures to assess an individual's level of PTSD symptoms, self-compassion, and PTG. The use of self-report questionnaires may have led participants to answer questions in a socially desirable manner.

Therefore, by using more extensive forms of measurement (i.e., behavioral observations, self- and other-report measures, etc.), this may provide a more accurate and objective method of measuring these constructs to ultimately confirm the relationship among variables found in the current study. Alternatively, multiple measures could be used to validate each construct and assess these pathways via structural equation modeling. Lastly, the current study found that self-compassion partially mediated the relationship between PTSD symptoms and PTG, suggesting that there are constructs other than self-compassion that may play an integral role in this relationship. Although the mediation model provides implications about causality, the current study utilized cross-sectional analyses six months following Hurricane Harvey. Thus, future longitudinal studies would be necessary to test the implications displayed in the current study. Longitudinal studies would allow one to see the potential curvilinear relationship and time course for the development of PTG in a trauma natural disaster population.

In conclusion, although there is a growing body of research focusing on the virtue of self-compassion on trauma research, little has been done to identify the relationship between PTSD symptoms, PTG, and self-compassion and the implications of such relationships. The current study aimed to better understand this relationship through a mediation analysis, in hopes that future research will test the benefits of using self-compassion based interventions to increase patient care and well-being. Findings from such research can help inform natural disaster trainings that target clinical effectiveness and protection against negative consequences of natural disaster trauma including PTSD. Further analyses looking at other mediation pathways in this relationship can further elucidate the unique relationship of self-compassion as a resilience factor for growth after trauma.

**Author Contributions:** Conceptualization, J.Y., D.C.W. and A.C.; methodology, J.Y. and D.C.W.; data curation, J.S. and D.C.W.; writing—original draft preparation, J.Y., D.C.W., A.C. and J.S. All authors have read and agreed to the published version of the manuscript.

**Funding:** This research received no external funding.

**Institutional Review Board Statement:** The study was approved by the Institutional Review Board of the University of Houston protocol code STUDY00000694 on 27 February 2018.

**Informed Consent Statement:** Informed consent was obtained from all subjects involved in the study.

**Data Availability Statement:** The data presented in this study are available on request from the corresponding author. The data are not publicly available due to limitations outlined by our Institutional Review Board, who approved the data collection process of this study.

**Conflicts of Interest:** The authors declare no conflict of interest.

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
