# Peer review of "Growth after Trauma: The Role of Self-Compassion following Hurricane Harvey"

_traumacare, doi:10.3390/traumacare1020011_

Round 1
Reviewer 1 Report
A very interesting article that examines the relationship between PTSD symptoms, PTG, and self-compassion, with self-compassion as a potential mediator.
I just have one point for clarification:
Looking at the regression, it is unclear why race was not included as a covariate, alongside age and gender, even though data for race was collected as indicated in the descriptive analysis.
Author Response
|
Thank you for reviewing our manuscript! We appreciate your point of clarification regarding the regression analysis through Process Macro in SPSS. You have a very fair point including ethnicity as a covariate. We have subsequently re-run our analysis including ethnicity. While controlling for these covariates, the mediation effects continue to hold. Accordingly, we have adjusted the results section and tables to address these concerns. |
Reviewer 2 Report
Hello,
I have made some suggestions for changes in the attached. The main paragraph that needs work is the one on PTG.

Author Response
We absolutely appreciate your review of our paper and providing constructional feedback to help us improve our manuscript.
With regard to all grammatical changes and errors found within your pdf, they have been fixed in our paper. More specifically with our PTG section, we have also amended and appropriately addressed your concerns to improve wording and clarity. We have also revised this section to reduce redundancy.
Round 2
Reviewer 2 Report
I think the changes have been made well. This a good and useful paper. There is one further change needed: the verb "hinders" should be "hinder">